# AMBRA1 p.Gln30Arg Mutation, Identified in a Cowden Syndrome Family, Exhibits Hyperproliferative Potential in hTERT-RPE1 Cells

**DOI:** 10.3390/ijms231911124

**Published:** 2022-09-22

**Authors:** Sundaramoorthy Revathidevi, Kazuyoshi Hosomichi, Toyoaki Natsume, Hirofumi Nakaoka, Naoko T. Fujito, Hisako Akatsuka, Takehito Sato, Arasambattu Kannan Munirajan, Ituro Inoue

**Affiliations:** 1Human Genetics Laboratory, National Institute of Genetics, Mishima 411-8540, Japan; 2Laboratory of Computational Genomics, Tokyo University of Pharmacy and Life Sciences, Hachioji 192-0392, Japan; 3Molecular Cell Engineering Laboratory, National Institute of Genetics, Mishima 411-8540, Japan; 4Research Center for Genome & Medical Sciences, Tokyo Metropolitan Institute of Medical Science, Tokyo 156-8506, Japan; 5Department of Cancer Genome Research, Sasaki Institute, Sasaki Foundation, Chiyoda 101-0062, Japan; 6Division of Host Defense Mechanism, Tokai University School of Medicine, Isehara 259-1193, Japan; 7Department of Genetics, Dr. ALM PG Institute of Basic Medical Sciences, University of Madras, Taramani, Chennai 600113, India

**Keywords:** *AMBRA1*, *AMBRA1* Q30R mutant, Cowden syndrome, primary cilia, CRISPR/Cas, G1/S transition

## Abstract

Cowden syndrome (CS) is a rare autosomal dominant disorder associated with multiple hamartomatous and neoplastic lesions in various organs. Most CS patients have been found to have germline mutations in the *PTEN* tumor suppressor. In the present study, we investigated the causative gene of CS in a family of *PTEN* (phosphatase and tensin homolog deleted on chromosome 10) -negative CS patients. Whole exome sequencing analysis revealed *AMBRA1* (Autophagy and Beclin 1 Regulator 1) as a novel candidate gene harboring two germline variants: p.Gln30Arg (Q30R) and p.Arg1195Ser (R1195S). *AMBRA1* is a key regulator of the autophagy signaling network and a tumor suppressor. To functionally validate the role of *AMBRA1* in the clinical manifestations of CS, we generated *AMBRA1* depletion and Q30R mutation in hTERT-RPE1 (humanTelomerase Reverse Transcriptase-immortalized Retinal Pigmented Epithelial cells) using the CRISPR-Cas9 gene editing system. We observed that both *AMBRA1*-depleted and mutant cells showed accumulation in the S phase, leading to hyperproliferation, which is a characteristic of hamartomatous lesions. Specifically, the *AMBRA1* Q30R mutation disturbed the G1/S transition of cells, leading to continuous mitotic entry of mutant cells, irrespective of the extracellular condition. From our analysis of primary ciliogenesis in these cells, we speculated that the mitotic entry of *AMBRA1* Q30R mutants could be due to non-functional primary cilia that lead to impaired processing of extracellular sensory signals. Additionally, we observed a *situs inversus* phenotype in *ambra1*-depleted zebrafish, a developmental abnormality resulting from dysregulated primary ciliogenesis. Taken together, we established that the *AMBRA1* Q30R mutation that we observed in CS patients might play an important role in inducing the hyperproliferative potential of cells through regulating primary ciliogenesis.

## 1. Introduction

Cowden syndrome (CS), also known as multiple hamartoma syndrome, is a rare genetic autosomal dominant disorder affecting 1 in 200,000 people [1,2]. This syndrome is characterized by multiple hamartomatous and neoplastic lesions in various tissues and elevates the lifetime risk for several cancers [3,4]. The most common etiology of CS is germline mutations in PTEN, which have been reported in ~80% of CS patients [5]. In recent years, other CS predisposition genes such as KILLN (Killin), SDHB (Succinate Dehydrogenase Complex Iron Sulfur Subunit B), SDHC (Succinate Dehydrogenase Complex Subunit C), SDHD (Succinate Dehydrogenase Complex Subunit D), PIK3CA (Phosphatidylinositol-4,5-Bisphosphate 3-Kinase Catalytic Subunit Alpha), AKT1 (AKT serine/threonine kinase 1), and SEC23B (SEC23 Homolog B, COPII Coat Complex Component) have been identified in CS patients [3]. However, for a significant proportion of individuals meeting the clinical diagnostic criteria for CS, the genetic basis of the disease is yet to be established. 

We investigated the causative gene of CS in a family: the father and two daughters were diagnosed with CS, while the mother and one daughter were not affected. We performed whole exome sequencing (WES) analysis on all members of the family and identified AMBRA1 as the top candidate. AMBRA1 could be a good candidate from the genetic results, but its functional significance needs to be examined.

AMBRA1, a key adaptor protein of the autophagy signaling network, is a tumor suppressor and mutated in a wide range of tumors [6,7]. In addition, depletion of AMBRA1 protein was shown to be associated with an enhanced rate of proliferation [8]. The mutations and functional deficiency of AMBRA1 have previously been reported to be associated with neural tube defects [9] and cancer [6]. However, AMBRA1 has not been reported in CS and CS-related diseases. Therefore, we evaluated the functional significance of the identified AMBRA1 germline variant Q30R and its phenotypic relevance to CS using cell line and zebrafish models. 

In our study, AMBRA1 Q30R mutant hTERT-RPE1 cells showed hyperproliferation, disturbed G1/S transition, and abnormal primary ciliogenesis. Primary cilia are immotile extracellular sensory organelles that are dynamically regulated during cell cycle progression. Primary cilia assemble in non-dividing cells of the G0/G1 phase and disassemble upon cell cycle entry [10]. Therefore, we found that AMBRA1 Q30R mutant protein might disrupt the G1/S cell cycle checkpoint through the “AMBRA1—primary cilia regulatory node”, thereby inducing hyperproliferation.

## 2. Results

### 2.1. Study Participants

The IRUD (Initiative on Rare and Undiagnosed Diseases) and J-RDMM (Japan Rare Disease Models and Mechanisms Network) are ongoing multi-center research projects supported by the Japan Agency for Medical Research and Development (https://j-rdmm.org/; accessed on 29 November 2018). As part of these projects, we aimed to identify the candidate gene for one CS family. The affected individuals of this study were the father and two daughters of the family (Figure 1A): the father (44 years old), proband, was diagnosed with CS, having benign polyposis in the colon, skin, and blood vessels based on the criteria of the International Cowden Consortium. One daughter (21 years old) had resection surgeries of the thyroid at the ages of 3 (left lobe) and 18 (right lobe) due to a diagnosis of Plummer disease. The other daughter also had surgery of the thyroid at the age of 17 (left lobe) and was diagnosed with Plummer disease. All three affected individuals did not harbor germline mutations in PTEN, which represent the common etiology of CS, or in any of the previously reported CS-related genes. We performed whole exome sequencing using genomic DNA of both affected (CS1, CS2, and CS3) and unaffected members of the CS family (Figure 1A).

### 2.2. AMBRA1 Is a Candidate Gene of Cowden Syndrome

From the WES data of the CS family, we identified nearly 19,000 germline variants each in both CS-affected and unaffected family members; all of them were in exonic/splicing sites that could potentially alter the structure or function of the proteins. We filtered out synonymous variants and variants which were previously reported in dbSNP or the 1000 Genomes Project. To identify the novel variants consistent with autosomal dominant inheritance, we considered rare variants that were heterozygous and present in all the affected individuals but absent in the unaffected family members (Figure 1B). Among the 17 genes harboring a total of 18 heterozygous variants (Figure 1C), we focused on AMBRA1 as the most likely candidate for CS in the family, and as the most functionally relevant because of its tumor suppressor function [6,7].

AMBRA1 harbored two germline variants: Q30R (c.A89G) and R1195S (c.G3585C). Q30R, an AMBRA1 variant, exhibited the most deleterious effect according to the in silico pathogenicity prediction (Figure 1C), located in the highly conserved N-terminal region of AMBRA1 (Figure 1D), which acts as a scaffold for several protein–protein interactions. Meanwhile, the other variant, R1195S, is located in the less conserved C-terminal domain of AMBRA1. Based on the conservation and functional significance of the N-terminal region of AMBRA1 and the highest pathogenic score of Q30R, we prioritized AMBRA1 Q30R as a potential candidate variant for further analysis to understand the impact on AMBRA1 function leading to the phenotypic changes observed in CS syndrome. 

### 2.3. AMBRA1 Dynamically Altered the Cell Cycle in hTERT-RPE1 Cells

#### 2.3.1. AMBRA1 Depletion Caused Hyperproliferation

Considering uncontrolled cell proliferation as a hamartoma characteristic and AMBRA1 as a newly recognized tumor suppressor, we speculated that AMBRA1 depletion might affect the cell proliferation rate. We created AMBRA1-depleted hTERT-RPE1 cells using the CRISPR-Cas9 system by deleting approximately 1093 bp between intron 6 and exon 7 (Appendix A) that could abolish AMBRA1 protein expression in the homozygous knockouts (AMBRA1^−/−^) (Appendix A). When we investigated the proliferation rate using Ki67 staining, we observed that the percentage of both heterozygous and homozygous AMBRA1 knockout hTERT-RPE1 cells (AMBRA1^+/−^ and AMBRA1^−/−^, respectively) entering the S and G2/M phases was significantly higher than that of the wildtype (Figure 2A,B), indicating that AMBRA1 deficiency accelerated the mitotic division. To confirm the accelerated G1/S phase transition of AMBRA1 knockouts, we analyzed the proliferation rate of cells after 48 h of serum starvation, which, in general, induces the cells to synchronize at the G1 phase, and inhibits cell division. In such nutrient-starved conditions, the cell cycle of both AMBRA1^+/−^ and AMBRA1^−/−^ knockout cells was arrested at G1, showing limited proliferation, similar to the wildtype cells (Figure 2C).

#### 2.3.2. AMBRA1 Q30R Mutation Disturbed the G1/S Transition of Cells

To examine whether the *AMBRA1* Q30R mutation affects cell proliferation similar to *AMBRA1* depletion, we introduced the Q30R mutation in exon 2 of *AMBRA1* to hTERT-RPE1 cells using the CRISPR-Cas9 gene editing system (Appendix A). We represented heterozygous and homozygous mutants as AMBRA1^Q30R A/G^ and AMBRA1^Q30R G/G^, respectively, where A/G and G/G represents the nucleotide change. We observed that the AMBRA1^Q30R A/G^ and AMBRA1^Q30R G/G^ mutants also showed increased proliferation by exhibiting significantly higher entry into the S and G2/M phases than the wildtype cells (Figure 2A,B). However, in nutrient-starved conditions, unlike the *AMBRA1* knockouts, AMBRA1^Q30R A/G^ and ^Q30R G/G^ cells continued to accumulate in the S/G2+M phase (Figure 2A,C). Our results demonstrate that both the *AMBRA1* knockout and Q30R mutation altered cell proliferation by inducing the accumulation of S-phase cells. The increased proliferation induced by the *AMBRA1* Q30R mutation could not be altered by nutrient deprivation, indicating that *AMBRA1* Q30R disturbed the G1/S transition of cells, irrespective of the extracellular condition. 

#### 2.3.3. AMBRA1 Knockouts and Q30R Mutants Exhibited Different Cell Cycle Behaviors

In nutrient-sufficient conditions, along with a higher proportion of cells in the S+G2/M phase, heterozygous AMBRA1^+/−^ showed a considerable proportion in the G0 phase, like the wildtype cells, whereas AMBRA1^−/−^, AMBRA1^Q30R A/G^, and AMBRA1^Q30R G/G^ cells showed a small proportion in the G0 phase (Figure 2D). Upon starvation, both wildtype and AMBRA1^+/−^ cells accumulated in the quiescent G0 stage, while a larger percentage of AMBRA1^−/−^ cells were retained in the G1 phase (Figure 2E). Meanwhile, the AMBRA1^Q30R A/G^ and AMBRA1^Q30R G/G^ mutants showed a significant number of cells in the S/G2+M phase along with the G1 phase (Figure 2E). These observations suggest that AMBRA1 perturbation resulted in dysregulation of the dynamic balance between cell cycle arrest and quiescence in unfavorable extracellular conditions, which is otherwise maintained by functionally intact AMBRA1. 

### 2.4. AMBRA1 Depletion Affected Primary Ciliogenesis, the Structural Checkpoint of the Cell Cycle

#### 2.4.1. AMBRA1 Q30R Mutant hTERT-RPE1 Cells Exhibited Abnormal Primary Cilia

To elucidate the molecular mechanism of AMBRA1 controlling G0/G1 cell cycle arrest, we focused on the formation of primary cilia, in line with the fact that most cells begin to disassemble their primary cilia upon cell cycle re-entry [11]. We induced primary cilia formation by starving cells at high confluency for 24 h and quantified the frequency of ciliation in each cell type.

In nutrient-sufficient conditions, normal cells disassembled the primary cilia while entering into cell division (Figure 3A,B) and reassembled them upon serum starvation (Figure 3A,C), as seen in the wildtype cells. Upon starvation, the ciliation in the heterozygous knockout AMBRA1^+/−^ was numerically higher than that of the wildtype, without any structural changes. The homozygous knockout AMBRA1^−/−^ and the mutants AMBRA1^Q30R A/G^ and ^Q30R G/G^ showed numerically and structurally different primary cilia when compared to AMBRA1^+/−^ and wildtype cells, as described below. 

AMBRA1^−/−^ showed a reduced number of cilia compared to the wildtype cells (Figure 3C). This could coincide with the accumulation of more AMBRA1^−/−^ cells in the G1 phase and the entry of AMBRA1^+/−^ and wildtype cells into the quiescent G0 phase (Figure 2E). AMBRA1^Q30R A/G^ cells showed increased ciliation, albeit in nutrient-sufficient conditions, and abnormal primary ciliary structures upon starvation. The ciliary abnormalities included coiled, spiral, and disrupted ciliary structures (Figure 3D). Some AMBRA1^Q30R A/G^ cells even showed more than one cilium per cell (Figure 3D). All these abnormal ciliary structures might disrupt the function of primary cilia. On the contrary, homozygous AMBRA1^Q30R G/G^ cells did not form primary cilia in both nutrient-sufficient and starved conditions (Figure 3A–C). We speculated that the AMBRA1 Q30R mutant protein influenced the disassembly of primary cilia, showing that AMBRA1^Q30R A/G^ might disrupt the structural integrity of the primary cilia, while AMBRA1^Q30R G/G^ might govern the permanent mode of primary cilia disassembly. This correlated with the percentage of AMBRA1^Q30R A/G^ and AMBRA1^Q30R G/G^ cells entering the cell cycle.

The cells (wildtype and AMBRA1^+/−^) which entered the G0 phase (Figure 2E) during starvation exhibited normal ciliogenesis, while the cells (AMBRA1^−/−^, AMBRA1^Q30R A/G^, and AMBRA1^Q30R G/G^) which entered the cell division phases including G1 and S+G2/M (Figure 2E) showed ciliary abnormalities. Our results suggest that *AMBRA1* depletion and Q30R mutation dysregulated primary ciliogenesis, which further led to impaired processing of extracellular sensory signals, resulting in disrupted cell cycle checkpoints.

#### 2.4.2. In Vivo Knockout Model of ambra1 in Zebrafish Showed Situs Inversus Phenotype

Connecting our observation with the clinical manifestations of hamartomatous lesions in multiple organs of CS patients, we hypothesized that AMBRA1 depletion could affect multiple organs and show phenotypes associated with ciliogenesis and hyperproliferation in zebrafish. In zebrafish, two paralog nonredundant genes, ambra1a and ambra1b, have been observed [12]. At first, to investigate the function of ambra1 in zebrafish, we knocked out *ambra1a* and *ambra1b* in zebrafish with the CRISPR/Cas9 system. When both ambra1a and ambra1b were knocked out together, no specific phenotype was observed, except bleeding. However, when either *ambra1a* or *ambra1b* was knocked out independently, some CRISPANTs, especially ambra1a-depleted ones, frequently showed a *situs inversus* phenotype, a characteristic of ciliary abnormalities [13] (Figure 3E).

To confirm the association of ambra1a depletion with the *situs inversus* phenotype, we knocked down ambra1 in zebrafish using morpholinos and determined the subsequent phenotypic changes. We observed that ambra1a morphants showed significantly more situs inversus phenotypes than their wildtype counterparts (23.78%—34/143 and 6.62%—9/136, respectively) (Fisher exact *p* value < 0.0001) (Figure 3F). On the other hand, ambra1b morphants did not show any significant results. 

Our observation of the *situs inversus* phenotype in ambra1a morphants but not in ambra1b morphants suggests that these two genes play different roles in zebrafish embryogenesis, with ambra1a being involved in the orientation of visceral organs, which could plausibly be through regulating primary cilia. This confirms that AMBRA1 function is tightly correlated with ciliogenesis. However, we could not observe any lesion similar to a hamartoma in the ambra1a knockdown zebrafishes.

## 3. Discussion

In the present study, the overview of which is presented in Figure 4, we identified *AMBRA1* Q30R as a novel candidate germline mutation of Cowden syndrome. The Q30R germline mutation, located in the highly conserved N-terminal region of AMBRA1, has not been documented previously. The N-terminal region of AMBRA1 is characterized by three WD40 domains which act as protein interaction scaffolds mediating the interaction of AMBRA1 with a wide range of proteins [14]. Deletion of 1-43 amino acids of the N-terminal region has been reported to alter the binding of AMBRA1 with several adaptor proteins such as DDB1 (DNA Damage Binding protein 1) and ELOC (Elongin C), which resulted in both the loss and gain of protein interactions [15]. This signifies that the Q30R mutation, near this WD40 domain, could have an impact on AMBRA1 scaffold function. 

Our observation of increased proliferation in AMBRA1-perturbed cells (AMBRA1^+/−^, ^−/−^, ^Q30R A/G^, ^Q30R G/G^) confirmed the hyperproliferative potential of the *AMBRA1* Q30 mutation, which could possibly be responsible for the benign self-limited growth of hamartomatous lesions in CS patients [16,17]. In addition, we observed that the monoallelic AMBRA1^+/−^ cells apparently showed a higher proliferation rate than the wildtype cells, which is consistent with previous reports of increased tumorigenesis in monoallelic *AMBRA1* mouse embryonic fibroblast cells [7], suggesting *AMBRA1* as a haploinsufficient tumor suppressor, sensitive to the gene dosage.

However, in nutrient-deprived conditions, wildtype and AMBRA1^+/−^ cells were retained in the quiescent G0 stage, while AMBRA1^−/−^ cells were retained in the G1 phase, and the mutants AMBRA^Q30R A/G^ and ^Q30R G/G^ proceeded to mitotic division. From these results, it is evident that AMBRA1 played a distinguishing role between the quiescent G0 and proliferating G1 phases, regulating both G0/G1 transition (cell cycle re-entry) and G1/S transition. We suggest that AMBRA1^−/−^ cells could have lost G0/G1 transition control, while AMBRA1 Q30R mutant cells lost G1/S phase transition control. The mutants, AMBRA^Q30R A/G^ and ^Q30R G/G^, showing uncontrolled proliferation, albeit the unfavorable extracellular condition, signifies the pathogenicity and irreversible hyperproliferative ability of Q30R mutation and its possible influence on benign hamartoma and malignant transformation in CS patients.

Recently, it has been reported that AMBRA1 regulates the abundance of D-type cyclins and controls the transition from G1 to S phase [8,18,19]. Cyclin D1 is a positive regulator of S phase entry during cell division, and its increased expression and lower proteasomal degradation in AMBRA1-deficient conditions caused the cells to have a shorter G1 phase, with a faster entry and longer residence into the S phase [8]. To confirm whether *AMBRA1* depletion and *AMBRA1* Q30R mutation could induce hyperproliferation through regulating cyclinD1, we analyzed cyclinD1 expression in AMBRA1^+/−^, AMBRA1^−/−^, AMBRA^Q30R A/G^ and ^Q30R G/G^ cells (Appendix A). AMBRA1^−/−^ cells showed high expression of cyclinD1, both in nutrient-sufficient as well as deprived conditions, indicating that the loss of AMBRA1 protein severely affected the expression of cyclinD1. However, AMBRA^Q30R A/G^ and ^Q30R G/G^ cells did not show any significant change in cyclinD1 expression level, indicating that AMBRA1 Q30R mutation might influence cell proliferation through the cyclinD1-independent pathway. 

Our observation of perturbed ciliogenesis in AMBRA1^−/−^, AMBRA^Q30R A/G^, and AMBRA^Q30R G/G^ cells could add insights into the possible mechanism of how these cells bypass the G0/G1 phase. Primary cilia are immotile microtubule-based structures, functioning as specialized sensory antennae for cells to detect extracellular signals critical for cell proliferation and differentiation [20]. The life cycle of primary cilia is tightly coupled with the cell cycle; cilia assemble in non-dividing cells of the G0/G1 phase and disassemble upon cell cycle entry upon nutrition-deficient and sufficient conditions, respectively [10]. In nutrient-deficient conditions, the AMBRA1 mediated autophagy process regulates the formation of primary cilia, which in turn suppresses mTORC1 dependent cell proliferation [11]. Therefore, disruption of primary cilia by AMBRA1 Q30R mutant protein may possibly result in impaired mTORC1 downregulation that might make the mutant cells non-responsive to cell cycle arrest signals at G0/G1 phases in nutrient deprived condition. 

Additionally, we observed a non-linear relationship between *AMBRA1* Q30R genotypes (A/G, G/G) and the ciliogenesis phenotype. The heterozygous AMBRA1^Q30R A/G^ cells showed ciliary assembly in nutrient-adequate conditions, and an irregular structure and multiciliation upon nutrient deprivation, while on the contrary, the homozygous AMBRA1^Q30R G/G^ cells showed no ciliation. It is not uncommon for a single mutation in highly pleiotropic genes that are involved in more than one function and have multiple protein interactions to result in very different phenotypes depending upon the alleles involved [21,22]. AMBRA1 is an intrinsically disordered protein (IDP) that is characterized by long regions of protein sequences that can fold upon binding to their interaction partners, allowing different types of protein folding with different partners. Our prediction of protein structural change of AMBRA1 protein induced by *AMBRA1* Q30R and R1195S using AlphaFold2 demonstrated that both the mutations did not induce a significant structural change in the AMBRA1 tertiary structure (Appendix A). However, being an IDP with multiple protein interactions, the Arg^30^ of AMBRA1 could potentially harbor different interaction partners and exhibit different ciliogenesis phenotypes with varying heteromeric conformations.

In this study, we demonstrated that AMBRA1 depletion induced hyperproliferation of cells by disrupting primary cilia and could possibly be a novel candidate of Cowden syndrome. Considering the phenotypic manifestation of CS in multiple organs, we tried to demonstrate the impact of AMBRA1 depletion at the organ level using a zebrafish model. In our analysis, we could not observe hyperproliferation when we knocked down the endogenous *ambra1a* of zebrafish. Conversely, the most noticeable phenotype we observed in *ambra1a* morpholino zebrafish was *situs inversus*, which is a congenital condition involving a mirrored orientation of visceral organs and strongly associated with the dysfunction/loss of primary cilia in the embryonic node [13]. Our observation of *situs inversus* added further emphasis to the significant role of AMBRA1 in primary cilia regulation and congenital defects. Therefore, we speculated that the “*AMBRA1*—primary cilia regulatory node” is involved in diverse functions from development to malignancy depending on the cellular context and extracellular environment and could possibly be associated with many other congenital conditions, which needs to be further explored.

In humans, there have been a few reports of patients showing *situs inversus* along with hamartoma disorders such as hepatic angiomyolipoma, tuberous sclerosis complex, and hepatic tumors [23,24,25], but the underlying genetic cause has not been determined. Our observations of AMBRA1 mutations in CS patients exhibiting hyperproliferative potential and ambra1 morphants in zebrafish showing a *situs inversus* phenotype could indicate that *AMBRA1* may act as a key player in the occurrence/co-occurrence of these two rare genetic conditions. Thus, our study will pave the way for future research regarding *AMBRA1* as a potential candidate gene of rare genetic conditions and exploring its functional implications.

## 4. Materials and Methods

### 4.1. Identification of Candidate Gene of PTEN Wildtype CS

This study was approved by the Institutional Research Board of the National Institute of Genetics (NIG#26-3; approved on 22 May 2014). After participants gave informed consent, we performed whole exome sequencing using genomic DNA of both the affected and unaffected members of the CS family. The detailed procedure of WES is presented in the Supporting Information. We prioritized novel germline variants (i) based on their potential to alter protein structure or function (e.g., missense, stop gain/loss, splicing) and (ii) those that were not reported in dbSNP and the 1000 Genomes Project, (iii) excluding common variants found in both the affected and unaffected family members and including germline mutations observed in all three probands, (iv) based on PhyloP, Polymorphism Phenotyping v2 (Polyphen-2) pathogenicity scores and GERP conservation scores. We predicted the structural changes induced by the missense variants (p.Q30R and p.R1195S) of our candidate gene AMBRA1 using AlphaFold2. Detailed protocol of the 3D-structural analysis is presented in the Appendix A [26,27].

### 4.2. Functional Analysis of AMBRA1 Q30R Using hTERT-RPE1 Cells

hTERT-RPE1 cells were maintained in Dulbecco’s Modified Eagle’s Medium (DMEM)-low glucose supplemented with 10% heat-inactivated fetal bovine serum and 1% Penicillin–Streptomycin. We created AMBRA1-depleted hTERT-RPE1 cells using the CRISPR-Cas9 system by deleting approximately 1093 bp between intron 6 and exon 7 (Appendix A). For *AMBRA1* Q30R editing (c.A89G), we designed a guide RNA (Appendix A) targeting exon 2 and a single-stranded oligodeoxynucleotide (ssODN) carrying the A89G mutation to facilitate homology-directed repair at the Cas9 cut site. We used a Neon transfection system (Invitrogen, Waltham, MA, USA) to electroporate the CRISPR/Cas9-RNP complex into the cells. To confirm the AMBRA1 knockout, we isolated genomic DNA from the clones and amplified the region spanning intron 6 and exon 7 of *AMBRA1* to produce a fragment of 204 bp if there was a deletion or 1300 bp if the genomic region was intact and resolved it in 10% PAGE (Appendix A). The AMBRA1 protein was detected using Western blotting (Appendix A). To confirm the *AMBRA1* Q30R editing, we amplified the exon 2 region and performed Sanger sequencing (Appendix A). Details are provided in the Supporting Information. 

We analyzed the distribution of AMBRA1^+/−^ and AMBRA1^−/−^ knockouts and AMBRA1^Q30R A/G^ and AMBRA1^Q30R G/G^ mutants in the cell cycle phases through flow cytometry using a BD Accuri flow cytometer (BD Biosciences, Franklin Lakes, NJ, USA) with Ki67 staining using an anti-Ki67 antibody (ab8191, Abcam, Cambridge, UK) and DNA staining using PI. The fractions of cells in different cell cycle phases were calculated using FCS express software (BD Biosciences, Franklin Lakes, NJ, USA). 

We observed the frequency of primary cilia in AMBRA1^+/−^ and AMBRA1 ^−/−^ knockout and AMBRA1^Q30R A/G^ and AMBRA1^Q30R G/G^ mutant cells in both nutrient-sufficient and starved conditions through immunocytochemistry using rabbit anti-arl13B primary antibody (C827D13, Proteintech, Rosemont, IL, USA), a primary cilia marker. The immunostained cells were imaged under a Deltavision Personal DV fluorescent microscope (GE Healthcare, Chicago, IL, USA). Quantification analysis of the nucleus and primary cilia was performed using Volocity 6.5 software (Quorum technologies, Pushlinch, ON, Canada). 

Detailed protocols of the flow cytometry and immunocytochemistry are presented in the Appendix A.

### 4.3. Manipulation of ambra1 in Zebrafish

All zebrafish experiments were carried out with permission from the Committee for Animal Care and Use of the National Institute of Genetics, Japan (NIG#29-14, approved on 4 July 2017). Adult zebrafish were maintained on a 13 h light/11 h dark cycle. Embryos were kept at 28° until processing for phenotypic analysis. In zebrafish, *ambra1* codes for two paralog nonredundant genes, *ambra1a* and *ambra1b* [12]. We manipulated zebrafish ambra1a and b using CRISPR/Cas9 and morpholinos following the protocol described by Wu et al. and Ansai et al. [28,29]. Phenotypic changes in *ambra1* CRISPANTS and morphants were analyzed using a Leica M165 FC microscope (Wetzlar, Germany). Details of the CRISPR/Cas9 and morpholino experiments on zebrafish are provided in the Appendix A [28,29]. 

### 4.4. Statistical Analysis

Statistical analyses such as ANOVA to compare AMBRA1^+/−^, AMBRA1^−/−^, AMBRA1^Q30R A/G^, and AMBRA1^Q30R G/G^ with the AMBRA1 wildtype cells and Fisher’s exact test to compare the *situs inversus* phenotype of ambra1a morphants with the wildtype zebrafish were performed using GraphPad prism V.9. (Dotmatics, San Diego, CA, USA).

## Figures and Tables

**Figure 1 ijms-23-11124-f001:**
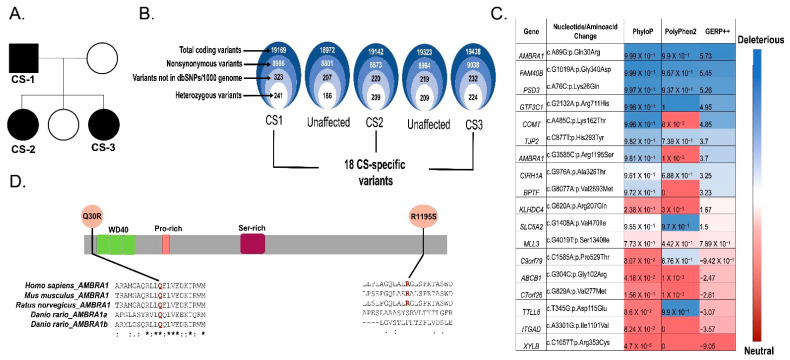
Identification of *AMBRA1* Q30R as a possible candidate germline variant using whole exome sequencing data of CS patients. (**A**) Pedigree of CS family comprising three affected (CS1, CS2, CS3) and two unaffected individuals. (**B**) Schematic representation of variant filtering and prioritization performed on WES data of the 3 CS probands and 2 unaffected family members. (**C**) In silico pathogenicity prediction and conservation score of 18 CS-specific variants. (**D**) Schematic representation of the AMBRA1 protein showing WD domains and CS variants Q30R locating near the WD40 (Tryptophan-Aspartic Acid repeats) domain and R1195S locating at the C-terminal region. Clustal-W2 Multiple sequence alignments at the bottom showed evolutionary conservation of the affected residues (highlighted in red) among species. The symbol * (asterisk) below the multiple sequence alignment indicates positions with a single, fully conserved residue; the symbol: (colon) indicates conservation between groups of strongly similar properties and the symbol. (period) indicates conservation between groups of weakly similar properties.

**Figure 2 ijms-23-11124-f002:**
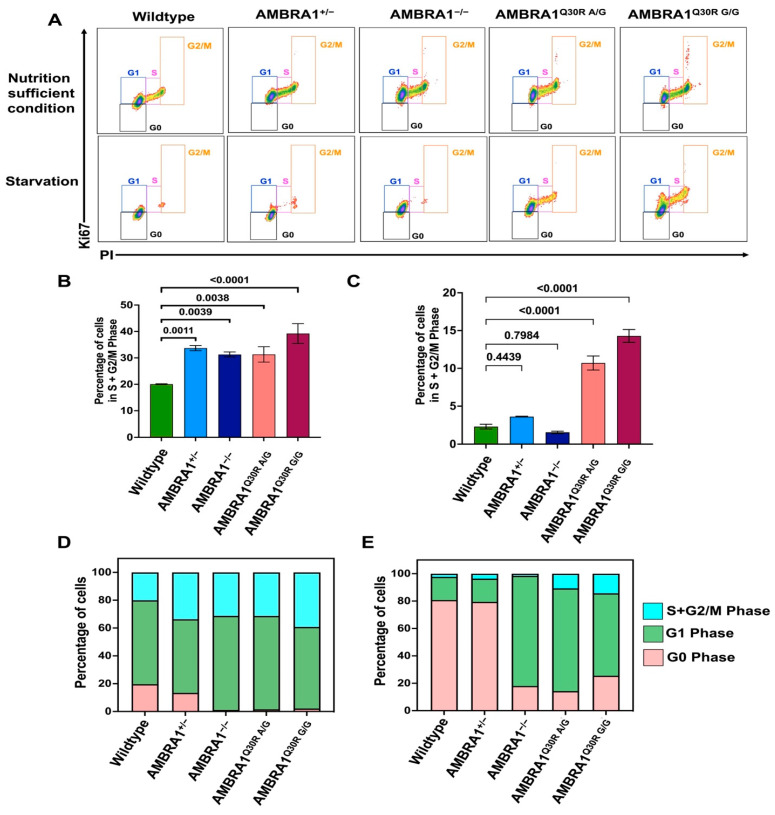
AMBRA1 deregulation dynamically altered the cell cycle arrest in hTERT-RPE1 cells. (**A**) Flow cytometry analysis of *AMBRA1*-altered hTERT-RPE1 cells. Cells were stained with Ki67, a cell proliferation marker rarely detected in the non-dividing G0 phase and highly expressed in proliferating cells (maximum in the G2 and early M phases), and PI, a DNA marker, staining proportionally to the DNA content of the cell. Staining cells with both Ki67 and PI successfully differentiated each cell cycle phase, represented in boxes of different colors (G0—black; G1—blue; S—pink; G2/M—orange). Analysis was performed in triplicate. A representative graph for each genotype is shown. (**B**) Bar graph representing the percentage of *AMBRA1*-depleted and Q30R mutant hTERT-RPE1 cells distributed in the S+G2/M phase under nutrient-sufficient conditions, calculated from the flow cytometry result (A, top). Statistical significance was determined by ANOVA. (**C**) Bar graph representing the percentage of starved *AMBRA1*-depleted and Q30R mutant hTERT-RPE1 cells distributed in the S+G2/M phase, calculated from the flow cytometry result ((**A**), bottom). Statistical significance was determined by ANOVA. (**D**) Stacked bar graph representing the percentage of *AMBRA1*-depleted and Q30R mutant hTERT-RPE1 cells distributed in different cell cycle phases under nutrient-sufficient conditions. (**E**) Stacked bar graph representing the percentage of starved *AMBRA1*-depleted and Q30R mutant hTERT-RPE1 cells distributed in different cell cycle phases.

**Figure 3 ijms-23-11124-f003:**
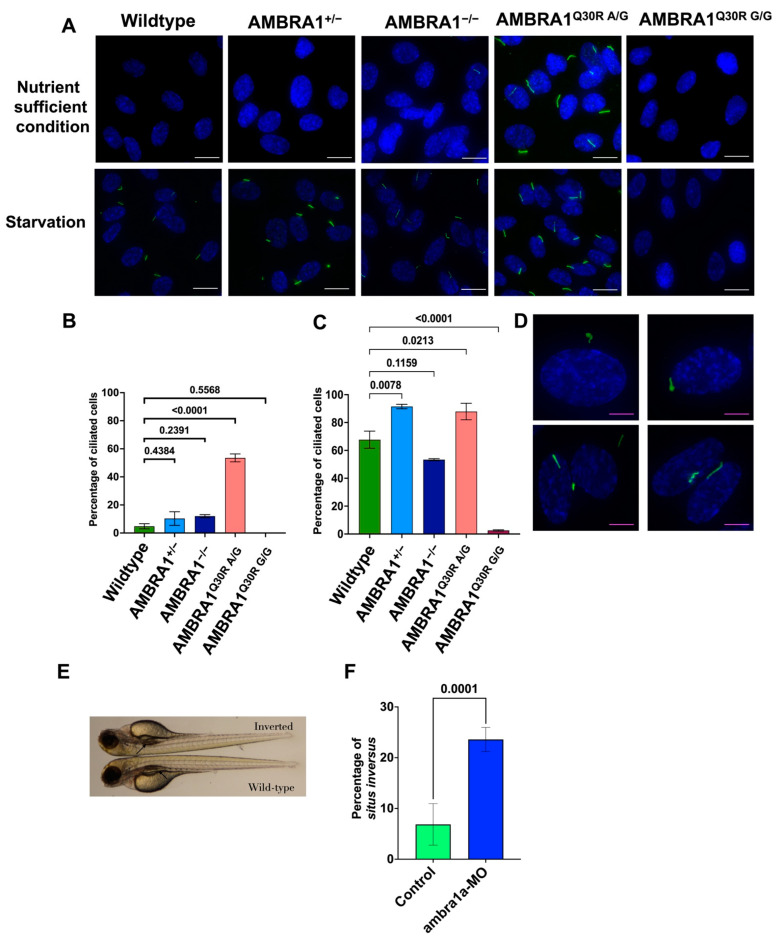
*AMBRA1* Q30R mutant hTERT-RPE1 cells exhibited abnormal primary ciliogenesis. (**A**) Immunocytochemistry images of cells stained with Arl13B (ADP Ribosylation Factor Like GTPase 13B), a primary cilia marker (stained green), and DAPI (4′,6-diamidino-2-phenylindole), a nucleus marker (stained blue). White scale bar = 25 μm (**B**) Bar graph representing the percentage of ciliated cells in each AMBRA1 genotype in nutrient-sufficient condition. Statistical significance was determined by ANOVA. (**C**) Bar graph representing the percentage of ciliated cells in each AMBRA1 genotype under starvation. Statistical significance was determined by ANOVA. (**D**) Representative immunocytochemistry images of AMBRA1^Q30R A/G^ mutant cells stained with Arl13B and DAPI, showing abnormal ciliary structures (curled—top; double cilia—bottom left; spiral—bottom right). Magenta scale bar = 15 μm (**E**) *ambra1a* zebrafish CRISPANT showing a *situs inversus* phenotype. The arrow indicates the reverse orientation of the gall bladder in *ambra1a* knockout zebrafish (top) and its normal orientation in wildtype zebrafish (bottom). (**F**) Bar graph representing the percentage of zebrafish showing *situs inversus* phenotype. Statistical significance was determined using Fisher’s exact test.

**Figure 4 ijms-23-11124-f004:**
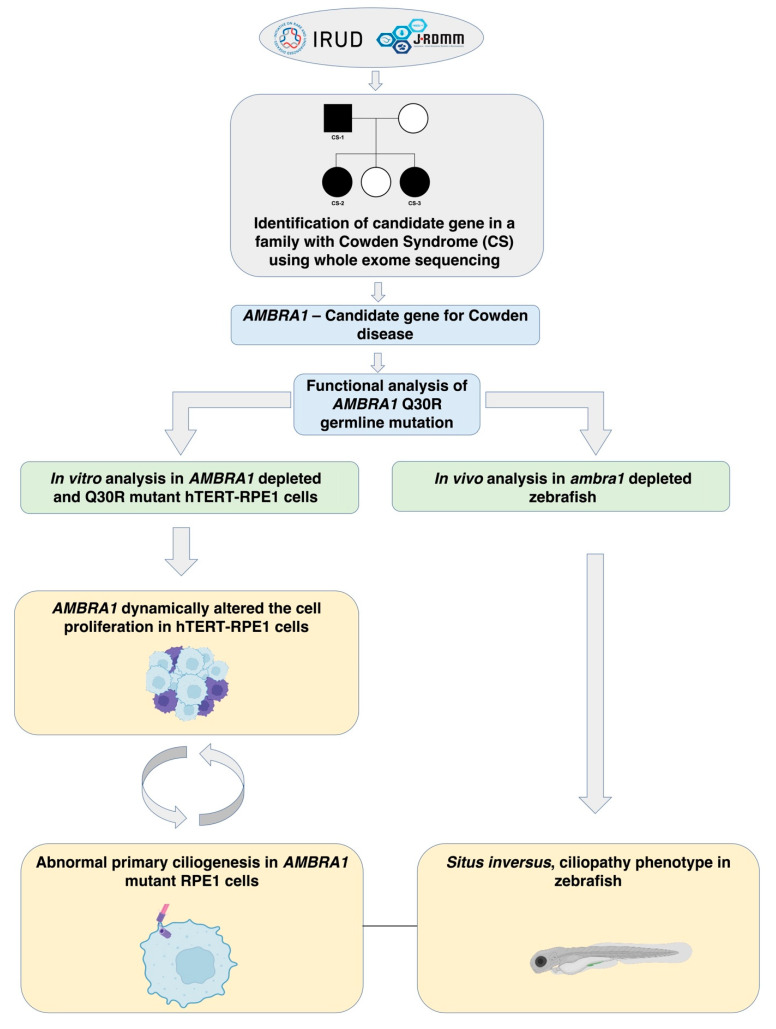
Flowchart for the identification and functional characterization of *AMBRA1* as a candidate gene of CS.

## Data Availability

Not applicable.

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
