# Peer review of "AMBRA1 p.Gln30Arg Mutation, Identified in a Cowden Syndrome Family, Exhibits Hyperproliferative Potential in hTERT-RPE1 Cells"

_ijms, 2022, doi:10.3390/ijms231911124_

Round 1

Reviewer 1 Report

Sundaramoorthy Revathidevi et al. report on "AMBRA1 Q30R mutation, identified in a Cowden syndrome 2 family, exhibits hyperproliferative potential in hTERT-RPE1 3 cells"

The data is interesting........

However, I have some queries.

1. Why patient clinical description is not given? Write a complete patent clinical description in the result part.

2. Write the pathogenicity score of the mutations identified.

3. 3D protein modeling should be performed for the identified missense variants.

4. Use 3 letter code for protein. Just visit the website for proper nomenclature. [https://varnomen.hgvs.org/].

5. A flowsheet diagram showing the whole story from the patient to the final findings will benefit the readers.

6. more zebrafish images should be added to compare the phenotypes if its overlapping with the patient phenotypes or not. Add in the discussion.

Reviewer 2 Report

The authors compared whole-exome sequences of Cowden syndrome patients and normal individuals from the same family and found that AMBRA1 contains two germline variants, Q30R (A89G) and R1195S (G3585C), in Cowden syndrome patients.  Since Q30R (A89G) is located in the highly conserved N-terminal region of AMBRA1, the authors further analyzed the effects of AMBRA1 deletion and AMBRA1(Q30A) in hTERT-RPE1 cells and the effect of ambra1 deletion in zebrafish.  It was found that AMBRA1 (Q30R) disturbed the G1-S transition of cells leading to continuous mitotic entry of mutant cells and affected the formation of primary ciliogenesis irrespective of the extracellular condition.  Moreover, an situs inversus phenotype was observed in ambra1-depleted zebrafish, while the developmental abnormality has been reported to result from dysregulated primary ciliogenesis.  Collectively, the authors suggest that AMBRA1 (Q30R) mutation observed in Cowden syndrome patients may play an important role in inducing hyperproliferative potential of cells by regulating primary ciliogenesis.  This study provides some interesting data, but lacks mechanism-based evidence.  Therefore, the manuscript should be revised further.

1.      The authors should determine the mechanism by which AMBRA1 regulates the formation of primary ciliogenesis.

2.      The authors should address the interacted partners of AMBRA1 in hTERT-RPE1 cells, and provide cyclin D levels in hTERT-RPE1 cells expressing AMBRA1 (Q30R) mutant.

3.      The authors should address how mutation of Q30R structurally affect the function of AMBRA1.  

Round 2

Reviewer 2 Report

I have no more comments.